# All-visible-light-driven salicylidene schiff-base-functionalized artificial molecular motors

Sven van Vliet[1,2,4], Jinyu Sheng [1,3,4], Charlotte N. Stindt[1] & Ben L. Feringa [1] ✉

Light-driven rotary molecular motors are among the most promising classes of responsive molecular machines and take advantage of their intrinsic chirality which governs unidirectional rotation. As a consequence of their dynamic function, they receive considerable interest in the areas of supramolecular chemistry, asymmetric catalysis and responsive materials. Among the emerging classes of responsive photochromic molecules, multistate first-generation molecular motors driven by benign visible light remain unexplored, which limits the exploitation of the full potential of these mechanical light-powered systems. Herein, we describe a series of all-visible-light-driven first-generation molecular motors based on the salicylidene Schiff base functionality. Remarkable redshifts up to 100 nm in absorption are achieved compared to conventional first-generation motor structures. Taking advantage of all-visible-light-driven multistate motor scaffolds, adaptive behaviour is found as well, and potential application in multistate photoluminescence is demonstrated. These functional visible-light-responsive motors will likely stimulate the design and synthesis of more sophisticated nanomachinery with a myriad of future applications in powering dynamic systems.

The impressive collection of molecular machinery found in Nature has served as a blueprint for scientists in their endeavour to control motion at the molecular level[1]. Artificial molecular machines enable controlled movement of their molecular components and facilitate the shift in contemporary chemistry from static molecules to dynamic and adaptive molecular systems and responsive materials[2-10]. Particularly attractive are the light-driven overcrowded-alkene-derived motors capable of performing repetitive unidirectional rotary motion upon illumination[11,12]. They have been exploited among others in liquid-crystals[13-15], adaptive catalysis[16,17], supramolecular systems[18-21] and porous materials[22-24]. The first generation of such overcrowded-alkene-based motors consists of identical upper and lower halves both featuring stereogenic centres which dictate the direction of rotation (Fig. 1). The second generation of motors possesses distinct motor halves in which a single stereogenic centre governs the

unidirectionality. For both classes of motors the 360° rotary cycle features two energetically uphill photoisomerization steps and two thermally activated, energetically downhill steps which are referred to as thermal helix inversions (THIs)[25].

In order to exploit the full potential of these light-driven molecular motors (MMs) in materials and biological systems, it is important to move away from the use of highly energetic and potentially damaging ultraviolet (UV) excitation, which is currently used to power several of these rotary motors. Various strategies have recently been explored to drive motor rotation by visible light, such as sensitization[26,27], metal complexation[28] or via bathochromically shifting the absorption maximum ($\lambda_{max}$) of the motor scaffold by virtue of electronic push-pull substituent effects[29,30] or π-system extension[31]. Furthermore, alternative motor designs, especially by the Dube group, responsive towards visible light have been described recently[32-34].

[1]Strathigh Institute for Chemistry, University of Groningen, Groningen, the Netherlands. [2]Present address: Department of Energy Conversion and Storage, Technical University of Denmark, Kgs, Lyngby, Denmark. [3]Present address: Institute of Science and Technology Austria, Klosterneuburg, Austria. [4]These authors contributed equally: Sven van Vliet, Jinyu Sheng. ✉e-mail: b.l.feringa@rug.nl

**A: Motor Scaffold**

**B: salicylideneaniline based switch**

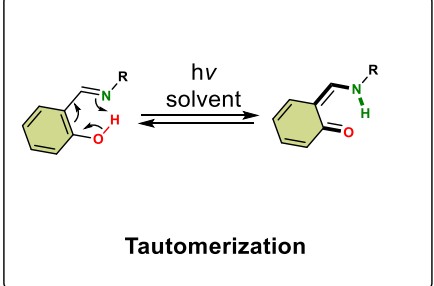

**C: This work: Salicylidene Schiff base molecular motors**

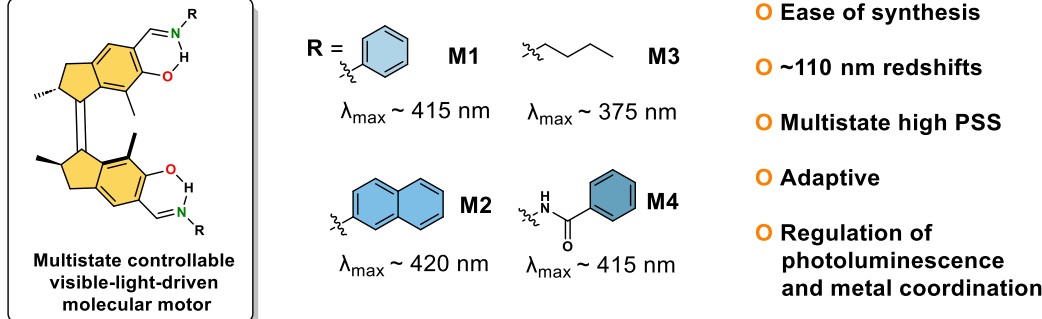

**Fig. 1 | The design of our work by merging a molecular motor core with a Salicylideneaniline motif. A** Conventional first-generation MM's unidirectional cycle triggered by 312 nm light. **B** Salicylideneaniline-based switch and its tautomerization behaviour. **C** The design of Salicylidene Schiff base functionalization towards visible-light-driven first-generation molecular motors with multiple advantages.

However, coupling visible-light addressability to applications within one simple motor design is intriguing, yet highly challenging[10]. So far, our efforts towards redshifting these systems have only been focussed on the second-generation MMs[35]. Recently, we demonstrated the synergistic combination of photoluminescence (PL) and motor rotation driven by visible light[36]. Surprisingly, the development of all-visible-light-powered first-generation MMs, which serve as unique multistate chiroptical switches[37,38], remains unreported. The majority of first-generation motor cores can only be driven by UV light (300–312 nm) (Fig. 1A)[39,40], severely limiting practical applications. Despite the absence of visible-light-driven first-generation MMs, these motors represent a prominent class of nanomachinery with intrinsically dynamic chirality which has been implemented in cholesteric liquid crystal materials, spin selective electron transport, and supramolecular systems to acquire spatiotemporal control over organization and properties in dynamic systems[10,41]. Whereas second-generation MMs are more suitable for continuous rotation applications, requiring rotary motion at velocious speed, first-generation MMs could fulfil a pivotal role in the growing demand for stimuli-adaptive multistage systems, capable of adopting multiple distinctive states with sequence specificity such as three-state chiroptical switches.

Salicylidene Schiff bases have drawn attention for decades due to their photochromic and thermochromic behaviour. Instalment of a salicylidene Schiff base unit onto a neighbouring arene motif leads in certain cases to strong absorption of visible light via a subtle interplay of the system's π-conjugation, the intramolecular H-bonding within the Schiff base framework and solvent polarity[42,43]. Besides potential visible-light absorption, the presence of salicylidene Schiff bases and derivatives offers additional prospects for coordination chemistry[44], (asymmetric) catalysis[45,46], supramolecular assembly[47], molecular photoswitches[48,49] and material sciences[50–52]. Recently, Zhu and coworkers developed a series of all-visible-light-responsive

dithienylethene (DTE) photoswitches via an intramolecular proton transfer (IPT) strategy governed by salicylidene Schiff base moieties, showing promising advantages of this functionality[53].

Herein, we report a general synthetic strategy to combine the first-generation MMs' modality with salicylidene Schiff base functionalities in one structure, thereby not only making the hybrid motor visible-light-responsive, but also amplifying distinctive physical features (e.g. photoluminescence (PL) and metal coordination affinity) of the different motor states. We devised all-visible-light-driven first-generation MMs with approximately 110 nm redshifted absorption maxima compared with the frequently used motor motifs upon instalment of salicylidene Schiff bases (Fig. 1). Furthermore, besides excellent photostationary state (PSS) distributions (i.e. high efficiency), varying the *N*-substitution is expected to affect the THI steps, thereby providing us with another tool for facile regulation of their rotary speed. Taking advantage of these all-visible-light-driven multistate motor scaffolds, the systems are employed in representative applications, including metal binding and photoluminescence modulation. We envision this simple synthetic modification to be an attractive strategy for the design and synthesis of future generations of visible-light-driven molecular motors.

## Results

### Motor synthesis

Taking advantage of our recently developed salicylaldehyde motor compounds[54], a one-step conversion using a Brønsted-acid-catalysed condensation protocol with aniline, 2-naphthylamine and *n*-butylamine leads the corresponding racemic bis(salicylimine) $Z_{st}$-**M1**-**M3** motors in good yields (Fig. 2A). By careful selection of the substituent attached to the imino nitrogen, we envisioned to tune the absorption maximum of the system upon extension of the π-system ($\lambda_{max}$ R = alkyl <R = phenyl <R = polycyclic arenes). $Z_{st}$-**M1**-OMe, in which the phenolic moieties of **M1** are methylated (Supplementary Information, section 2), was

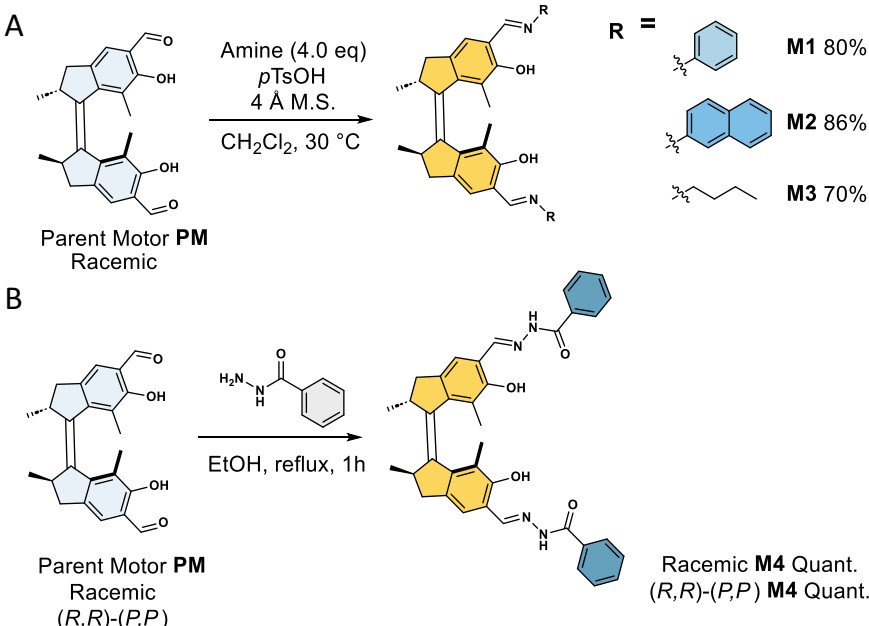

**Fig. 2 | Synthetic route to motors M1–M4 starting from the bis(salicylaldehyde) parent motor PM. A** Synthesis of racemic **M1** to **M3** by an imine condensation reaction. **B** Synthesis of racemic and enantiopure **M4** by a hydrazone condensation reaction.

prepared as a reference in order to investigate the potential effect of hydrogen bonding within the salicylidene Schiff base-based motif on the motor's (photochemical) properties. Beyond imines, the (acyl) hydrazone functional group has recently attracted attention in the fields of dynamic materials[55], metal coordination[56] and redox chemistry[57]. Elegant examples of the latter include hydrazone-based switches[58,59] and rotors[60] pioneered by Aprahamian and co-workers. To demonstrate the versatility of the formyl group as a handle for further functionalization of the motor scaffold, bis(salicaldehyde) motor **PM** was condensed with benzhydrazide to introduce the acylhydrazone functionality, which is closely related to the structural imine motif. Thus, reacting racemic and enantiopure (R,R)-(P,P) parent motor **PM** with benzhydrazide in boiling ethanol quantitatively gave racemic and (R,R)-(P,P) acylhydrazone-functionalized $Z_{st}$-**M4**, respectively (Fig. 2B). All motors were fully characterized by $^1$H NMR, $^{13}$C NMR, HRMS and Circular Dichroism (CD) spectroscopies. It is worth mentioning that motors **M1**–**M4** are very sensitive to ambient light, such that we would often observe slight amounts of isomerized compounds in $^1$H NMR, underlining indeed the visible-light responsiveness of the constructed motors by this strategy.

## Computational studies

Time-dependent density functional theory (TDDFT) is a well-established extension to density functional theory (DFT) to calculate properties for time-dependent systems, such as excitation energies and photo-absorption spectra[61]. TDDFT calculations at the CAM-B3LYP-D3BJ/def2-TZVPP/CPCM(CH₂Cl₂)//r²SCAN-3c/CPCM(CH₂Cl₂)[62–66] level of theory (Supplementary Information, section 12) showed a significant effect on the maximum absorption wavelengths upon instalment of salicylaldehyde Schiff bases in both halves of the first-generation motor scaffold (Table 1 and Supplementary Table 1). Compared to the conventional molecular motors lacking both the imine and hydroxy functionality (**M-oR**), the maximum absorption wavelengths of the hydrogen-bonded **M-OH** structures were calculated to be more than 50 nm redshifted. Furthermore, calculations show that the presence of only the hydroxy functionality (**M-oOH**) has a negligible effect on the absorption wavelength of these motors. Whereas instalment of only the imine functionality (**M-H**) already leads to an appreciable redshift, comparison of the salicylaldehyde Schiff base motors with their methoxy-substituted

analogues (**M-OMe**) shows that it is the unique combination of both the imine and hydroxy functionality that leads to the strongest bathochromic shift. For **M1** and **M2**, with aryl substituents at the Schiff base nitrogen, the calculated $\lambda_{max}$ corresponds to wavelengths >390 nm, less than 10 nm away from the 400 nm threshold, frequently considered as the onset of the visible light range for photochromic materials. For previous visible-light-driven second-generation motors, it has been shown that the predicted $\lambda_{max}$ is an underestimation of the experimental excitation wavelength[30], especially when solvent effects have a profound influence on the absorption properties. We therefore anticipate that these salicylaldehyde Schiff base incorporating first-generation motor systems can be fuelled with light at wavelengths beyond 400 nm (see experimental results, vide infra), thereby functioning as visible-light-responsive nanomachinery.

## Rotary behaviour of M1

Initial UV-Vis studies of imine-motor $Z_{st}$-**M1**, bearing a salicylidene Schiff-base substituent, and structurally related first-generation molecular **Motor-S1** (Fig. 3A) show that the absorption maxima are redshifted by >85 nm, with the absorption maximum of $Z_{st}$-**M1** in the visible range of the spectrum at *ca.* 415 nm (Fig. 2A). Although the $\lambda_{max}$ (385 nm) of $Z_{st}$-**PM** (Fig. 2) is already bathochromically shifted with respect to conventional first-generation motors, like **Motor-OH**, the redshift of $Z_{st}$-**M1** is nevertheless considerable with a difference > 30 nm between the absorption maxima of $Z_{st}$-**PM** and $Z_{st}$-**M1**.

To demonstrate that the operational cycle of **M1** is driven by visible-light irradiation, its unidirectional trajectory was confirmed by a combination of UV-Vis and $^1$H NMR spectroscopy (Fig. 3B–D). In accordance with our previously described first-generation molecular motors, the rotation cycle of **M1** involves two photochemical isomerization steps, each accompanied by an energetically downhill THI interconversion.

Upon irradiation of a solution of $Z_{st}$-**M1** (*cis* stable isomer, *i*-PrOH, 293 K) using 415 nm light, the characteristic absorption band of stable $Z_{st}$-**M1** at 400–425 nm decreased concomitantly with an increase in absorption at 425–475 nm. Due to the short half-life of metastable $E_{mst}$-**M1** (*trans* metastable isomer) at 20 °C, the appearance of the absorption band at 425–475 nm is indicative of the selective photochemical interconversion from stable $Z_{st}$-**M1** to metastable $Z_{mst}$-**M1** (*cis*

**Table 1 | Computed absorption wavelengths corresponding to the first transition of the stable Z conformations of the salicylidene Schiff-base-derived molecular motors and their related structures**

| | M-oR | M-oOH | M-H | M-OMe | M-OH |
|---|---|---|---|---|---|
| M1 | 336 | 332 | 379 | 384 | 395 |
| M2 | 347 | 342 | 385 | 389 | 407 |
| M3 | 308 | 316 | 349 | 353 | 360 |
| M4 | 338 | 339 | 379 | 384 | 387 |

Calculations were performed at the CAM-B3LYP-D3BJ/def2-TZVPP/CPCM(CH$_2$Cl$_2$)//r²SCAN-3c/CPCM(CH$_2$Cl$_2$) level of theory.

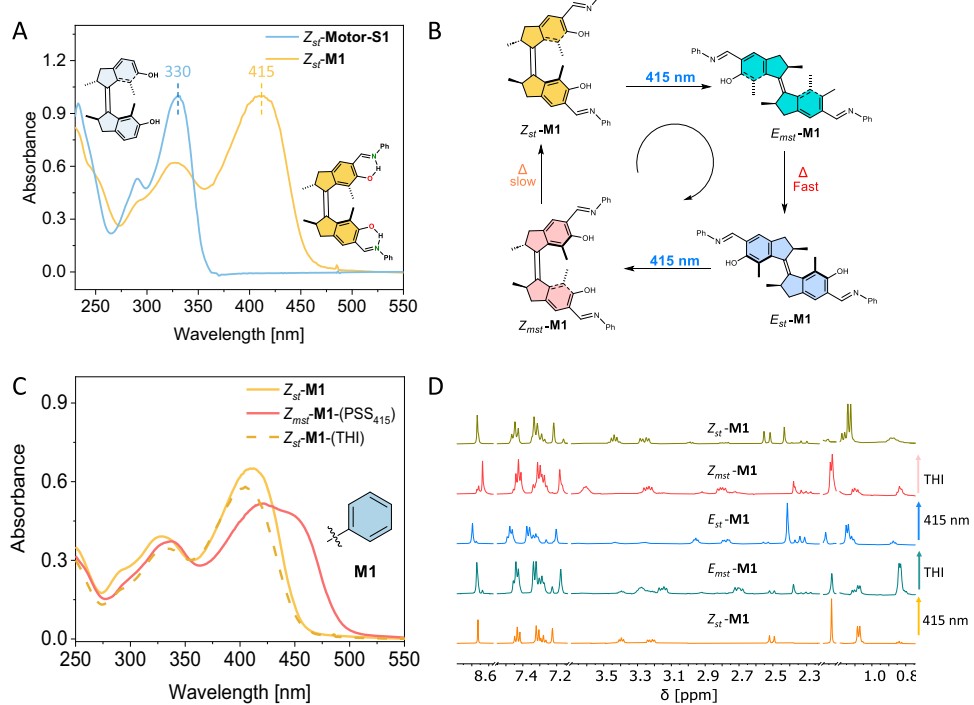

**Fig. 3 | Unidirectional rotary behaviour of motor M1. A.** Normalized UV-Vis spectra of $Z_{st}$-**M1** and motor motif $Z_{st}$-**Motor-S1**. **B** Schematic overview of the rotary cycle of **M1**. **C** UV-Vis analysis of the rotary cycle: before irradiation (orange), upon 415 nm light irradiation (red) and after THI (green). **D** $^1$H NMR (CD$_2$Cl$_2$, 600 MHz) spectra indicating rotary behaviour of **M1**: before irradiation ($Z_{st}$, yellow, bottom), PSS 415 nm at −40 °C ($E_{mst}$, cyan), THI ($E_{st}$, blue), PSS 415 nm at −40 °C ($Z_{mst}$, red) and THI ($Z_{st}$, green, top).

metastable isomer) through $E_{mst}$-**M1** and $E_{st}$-**M1**(*trans* stable isomer) intermediates (Fig. 3B and Supplementary Fig. 9). Subsequently, upon heating the sample at 50 °C, the previously formed absorption band at 425–475 nm ($\lambda_{max}$ = 423 nm, Fig. 2C) diminished while the initial absorption band at 400–425 nm was recovered, proving the THI process from $Z_{mst}$-**M1** to $Z_{st}$-**M1** to occur. Eyring analysis was performed to determine the Gibbs energy of activation for the THI process $Z_{mst}$-**M1** → $Z_{st}$-**M1** in order to calculate the thermal half-life ($t_{1/2}$) at room temperature which was found to be approximately 29 h. (Table 2 and Supplementary Fig. 4).

Further confirmation of the 360°rotary cycle and determination of the photostationary state (PSS) ratio of each photochemical step, i.e. $Z_{st}$-**M1** → $E_{mst}$-**M1** → $E_{st}$-**M1**→ $Z_{mst}$-**M1** → $Z_{st}$-**M1**, was provided by $^1$H NMR spectroscopy (CD$_2$Cl$_2$, 500 MHz) which allowed for the identification of all diastereoisomers following a unidirectional trajectory (Fig. 2D). Irradiation of stable $Z_{st}$-**M1** with 415 nm light at low temperature (−40 °C) led to formation of metastable $E_{mst}$-**M1**, resulting in a high PSS ratio, i.e., $Z_{st}$-**M1**: $E_{mst}$-**M1** = 81:19, with the characteristic signal at 0.8 ppm of $E_{mst}$-**M1**, thereby confirming that instalment of the sali-cylidene Schiff base functionalities does not compromise the light-induced isomerization performance of the motor. Allowing the sample to reach 25 °C gave quantitative conversion of metastable $E_{mst}$-**M1**

towards stable $E_{st}$-**M1** through the first THI pathway, thereby completing the first half of the rotary cycle. Consecutive illumination of $E_{st}$-**M1** at −40 °C resulted in photoisomerization towards metastable $Z_{mst}$-**M1** ($E_{st}$-**M1**: $Z_{mst}$-**M1** = 85:15 at the PSS) which was interconverted at room temperature to the initial diastereoisomer $Z_{st}$-**M1** via the second THI interconversion, thus completing the 360° rotation cycle (Fig. 3C, D). (For further support using CD spectroscopy of the unidirectional cycle of closely related Schiff base motor **M4**, see below).

### Impact of solvents and temperature on motor rotation behaviour

Due to the well-established polarity and temperature dependence of the tautomerization of salicylidene Schiff bases (Fig. 1B)[42,43], we studied the dependence of the absorption properties of $Z_{mst}$-**M1** as a function of solvent (Fig. 4A). UV-Vis spectra were measured in a wide range of solvents including protic and aprotic solvents (Fig. 4B and Supplementary Fig. 8). Indeed solvatochromism occurred in the form of redshifted bands in the more polar protic solvents (*i.e.* EtOH and MeOH) in accordance with the observation by Zhu[53] and London[67] for dithienylethene-based systems containing salicylidene Schiff bases. New bands emerge in polar protic solvents (Fig. 4B, orange and green spectra) which we attribute to the NH tautomer (keto form) of the salicylidene Schiff base induced via intramolecular proton transfer, implying an eminent influence on the photophysical properties of **M1**. TDDFT calculations indicate that the keto tautomers of motors **M1-M3** are indeed considerably redshifted compared with the enol tautomers (see Supplementary Table 1).

To investigate the effect of hydrogen bonding/tautomerization behaviour on the absorption properties of **M1** in greater detail, $Z_{st}$-**M1**-OMe was synthesized as a reference compound. **M1** and **Motor-S2** show characteristic absorption bands at $\lambda_{max}$ = 420 nm and $\lambda_{max}$ = 380 nm in isopropanol, respectively. This indicates that the bathochromic shift observed for **M1** is not exclusively an effect of the extension of the motor's π-system (Supplementary Fig. 8), but also a

**Table 2 | Gibbs free energy of activation (Δ‡G) and corresponding thermal half-lives ($t_{1/2}$) of helix inversion $Z_{mst}$ → $Z_{st}$ of motors M1–M4**

| Motor | Δ‡G($Z_{mst}$→$Z_{st}$)[a] [kJ mol$^{-1}$] | $t_{1/2}$[b] [h] |
|---|---|---|
| M1 | 100.8 ± 0.2 | 29.2 |
| M2 | 101.3 ± 0.7 | 35.7 |
| M3 | 100.0 ± 1.5 | 20.8 |
| M4 | 103.7 ± 1.4 | 97.9[c] |

[a]determined at various temperatures in *i*-PrOH, [b]defined as ln(2)/k at 20 °C, [c]obtained in DMSO.

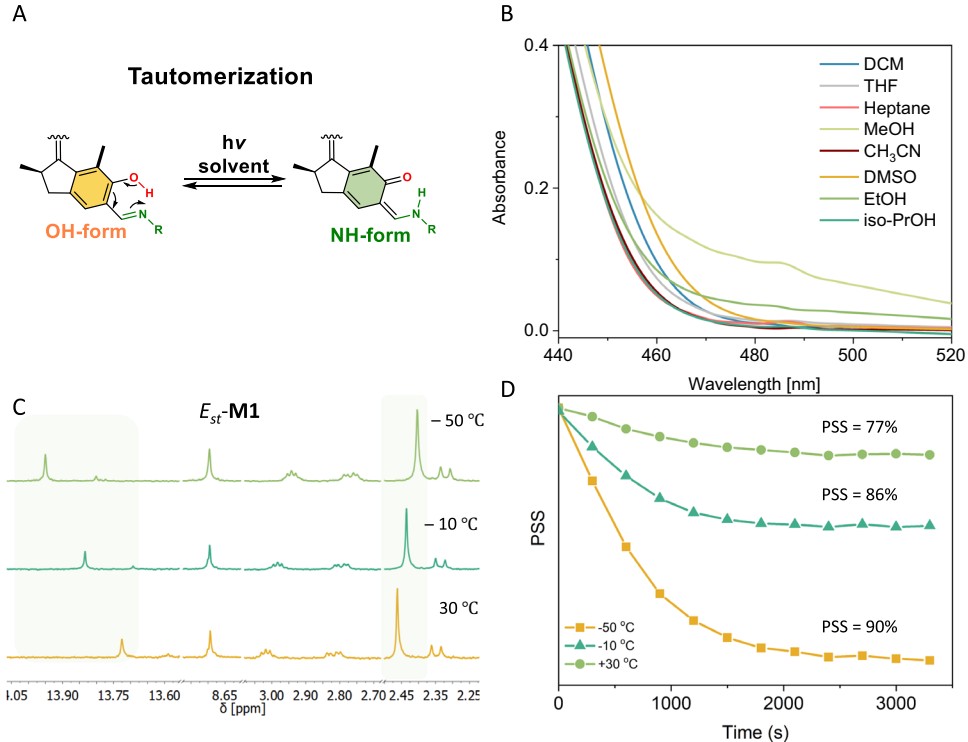

**Fig. 4 | Influence of solvent and temperature on the rotary behaviour of motor 1. A**. Illustration of tautomerization of **M1** between OH-form and NH-form. **B** Part of normalized UV-Vis spectra of $Z_{st}$-**M1** in a large set of solvents, close-up of the 440–520 nm window. **C** Chemical shifts of OH resonance in ¹H spectra of $E_{st}$-**M1** in CD₂Cl₂ at elevated temperatures, indicating the decreasing strength of H-bonding (from top to bottom). **D** Temperature dependence of the PSS from $E_{st}$-**M1** → $Z_{mst}$-**M1**.

consequence of the unique properties of the salicylidene unit, as was also confirmed by the TDDFT calculations (see Supplementary Table 1).

Remarkably, we further observed a solvent dependency on the rotation behaviour of **M1**. Motor **M1** displays high PSS ratios in polar, protic alcoholic solvents upon 415 nm light irradiation, while the PSS is sharply reduced in non-polar solvents (Supplementary Fig. 9), presumably due to the shift in the equilibrium between the OH- and NH-forms within the salicylidene Schiff base unit impacting photoisomerization of **M1**[68,69]. To support this explanation, we envisioned temperature as another factor affecting salicylidene Schiff base tautomerization and concomitantly motor rotation. When performing ¹H NMR irradiation studies at three distinctive temperatures, the PSS indeed showed to be temperature dependent with decreased PSS ratios at higher temperatures (Fig. 4D and Supplementary Fig. 13). The shifts of the OH resonance observed by ¹H NMR for $E_{st}$-**M1** at elevated temperatures indicate the difference in H-Bonding strength (Fig. 4C)[68]. Although the exact mechanism of this multiple stimuli dependency (i.e. solvent and temperature) on the PSS remains elusive and complicated, it is clear that the tautomerization process severely influenced the photoisomerization of the motors[70]. these findings might facilitate the exploration of smart, adaptable motor-based systems that can adjust to the conditions of their environment.

The incorporation of salicylidene Schiff-base moieties in these motor systems introduces the possibility of competing photochemical pathways, i.e., intramolecular excited-state proton transfer[71] and photoinduced $E$-$Z$ switching of the imine functionality[72], which could impart a reduction in the quantum yield of photochemical motor isomerization. However, the benefit that these motors possess, e.g., their visible-light responsiveness, outweighs the potential lowering of the photoisomerization quantum yields providing that the photostationary state (PSS) distributions and therefore the corresponding rotary functions remain largely unaffected. With respect to the latter, **M1** and **M2** reach PSS ratios well-over 80% at room temperature in non-polar solvents, fulfilling the key criteria, and reveal minimal effects of competing photochemical reaction channels for these specific motors.

## Extension of strategy to other salicylidene Schiff base derived motors

The successful strategy of introducing salicylidene Schiff bases in **M1** was used as a starting point for the construction of a collection of redshifted first-generation molecular motors. To illustrate, motor **M2**, containing naphthyl substituents at the imino nitrogens of the Schiff bases, exhibits a $\lambda_{max}$ at 420 nm, slightly more redshifted compared to **M1**, as a consequence of the extended π-system involving the naphthyl residues (Fig. 5A). Upon irradiation of stable $Z_{st}$-**M2** with 420 nm light at 20 °C, the characteristic absorption band located at 430 nm diminished, accompanied by the emergence of a new band at 460 nm, which can be ascribed to the photoinduced formation of metastable $Z_{mst}$-**M2**. ¹NMR spectroscopy revealed 90% of $Z_{mst}$-**M2** ($E_{st}$-**M2**: $Z_{mst}$-**M2** = 90:10) at the PSS, proving the excellent rotation behaviour (Supplementary Fig. 18). Eyring analysis at various temperatures was carried out to determine the Gibbs energy of activation for the THI step ($Z_{mst}$-**M2** → $Z_{st}$-**M2**) for the purpose of calculating the corresponding thermal half-life ($t_{1/2}$). The half-life of **M2** was found to be larger ($t_{1/2}$ = 36 h, 20 °C) compared to the value for **M1** (Table 2 and Supplementary Fig. 5).

The incorporation of an aliphatic butyl chain onto the imine functionalities in **M3** led to a $\lambda_{max}$ of stable $Z_{st}$ around 375 nm (Fig. 5B). It is noteworthy to mention that **M3** has no extension of its π-system beyond the imine residues of the salicylidene Schiff bases. While comparing **M3** to motor **M1**-OMe, which has an extended π-system, we observed that both motors show similar absorption maxima, in support of the additional bathochromic shifting effect due to the salicylidene Schiff base motif.

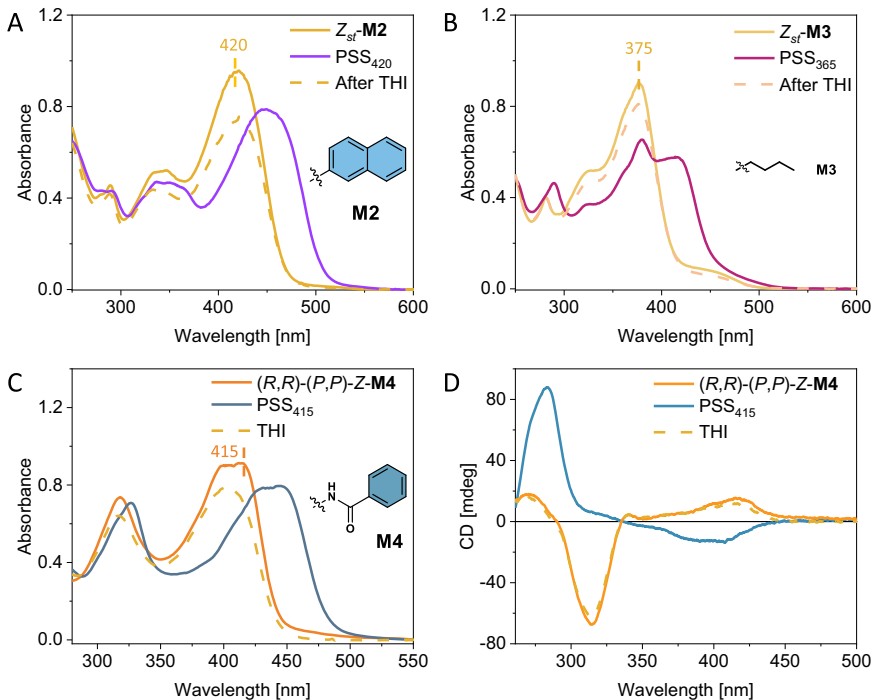

**Fig. 5 | Unidirectional rotary behaviour of motors M2 to M4. A**. UV-Vis spectra of $Z_{st}$-**M2** (orange), after 420 nm light irradiation (PSS, purple) and after THI (pink) measured in *i*-PrOH. **B** UV-Vis spectra of $Z_{st}$-**M3** (orange), after 365 nm light irradiation (PSS, purple-red) and after THI (cream) measured in *i*-PrOH. **C** UV-Vis spectra of $Z_{st}$-**M4** (orange), after 415 nm light irradiation (deep blue) and after THI (yellow) measured in DMSO. **D** CD spectra of $(R,R)$-$(P,P)$-$Z_{st}$-**M4** before irradiation (orange), after 415 nm light irradiation (PSS, blue), and after THI (yellow, dashed) measured in DMSO.

The full rotation cycles of **M2** and **M3** were studied in detail by a combination of UV-Vis and [1]H NMR spectroscopy. Consistent with the photochemical behaviour of **M1**, both **M2** and **M3** shows solvent and temperature dependence in reaching PSS. Remarkably, **M3** shows a much more significant temperature dependence of the photo-isomerization from $E_{st}$-**M3** to $Z_{mst}$-**M3** compared to **M1** and **M2**, as the PSS ratio decreased from 81:19 to 43:57 ($Z_{mst}$-**M3**:$E_{st}$-**M3**)[70]. To illustrate the versatility and tunability of the salicylidene Schiff base derived first-generation motor structural motif, we next installed a Schiff base alternative, the structural related acylhydrazone functionality, which yielded motor $Z_{st}$-**M4** as well as the enantiopure $(R,R)$-$(P,P)$ $Z_{st}$-**M4** via a simple synthetic protocol (Fig. 1B).

Like **M1** – **M3**, the main absorption band of **M4** undergoes an appreciable redshift with the $\lambda_{max}$ located at 405 nm, thus expanding the toolbox of first-generation motors with an additional scaffold responsive towards visible light (Fig. 5C). Upon irradiation with 415 nm light at 20 °C, stable $Z_{st}$-**M4** photochemically isomerized into metastable $E_{mst}$-**M4**, with clear isosbestic points. However, due to the short half-life of metastable $E_{mst}$-**M4** at ambient temperatures, the decrease of the absorption band around 390–410 nm and the appearance of the characteristic absorption band around 400–450 nm represents the selective photochemical interconversion between $Z_{st}$-**M4** to $Z_{mst}$-**M4**. The initial absorption band for $Z_{st}$-**M4** (390–410 nm) was obtained again after heating the sample in the dark at elevated temperatures, thereby permitting the THI to occur ($t_{1/2}$ = 98 h, Table 2 and Supplementary Fig. 7). To demonstrate fatigue resistance and overall stability, **M4** was subjected to multiple consecutive irradiation and heating cycles which were followed by [1]H NMR spectroscopy. Over the course of five rotary cycles, the initial and final spectra of $Z_{st}$-**M4** showed to be indistinguishable, confirming clean (photo)isomerization and general robustness of this motor system (Supplementary Fig. 26). The PL of **M1** can be reversibly modulated for several cycles without fatigue, indicating excellent and advantageous (photo)stability (Supplementary Fig. 25).

To unequivocally establish the unidirectional rotary cycle of **M4** was further analysed by CD spectroscopy (Fig. 5D and Supplementary Fig. 3). $(R,R)$-$(P,P)$ $Z_{st}$-**M4** exhibits a positive Cotton effect located at 425–340 nm in the CD spectrum. Upon irradiation at room temperature, the positive signal disappeared concomitantly with the emergence of a negative Cotton effect at 525–400 nm, characteristic of the opposite chirality of $(M,M)$-$Z_{mst}$-**M4**. A nearly identical CD spectrum was obtained after heating the same sample at elevated temperatures in the dark, indicating the progression of the THI from $(M,M)$-$Z_{mst}$-**M4** to $(P,P)$-$Z_{st}$-**M4**, thus completing the 360° rotation. Alternatively, irradiating a solution of $Z_{mst}$-**M4** solution at 455 nm led to the formation of $E_{st}$-**M4**, which results in a characteristic blue-shifted absorption band ($\lambda_{max}$ around 400 nm) and a concomitant change of the CD spectrum. It is worth noting that in the case for **M4** no temperature dependence on the photochemical behaviour was observed (Supplementary Fig. 17), which might be due to the stable structure of the acylhydrazone which possibly does not show a tautomerization process.

## Demonstration of potential applications as multistate photoswitches driven by visible light

The first-generation molecular motor core is a prominent scaffold known as a multistate photoswitch which has been widely used in controlling dynamic systems by light. Notably, the system is enabled to drive first-generation molecular motors as a multistate photoswitch solely by two distinct wavelengths of visible light (Fig. 6A), opening up new avenues to construct dynamic systems and materials.

Irradiation of $Z_{st}$-**M1** at room temperature by 415 nm light led to the direct formation of $Z_{mst}$-**M1** across ¾ rotary cycle (Fig. 6B, red line). Subsequently, $Z_{mst}$-**M1** can be selectively isomerized towards stable $E_{st}$-**M1** at 455 (Fig. 6B, blue line), thereby making **M1** an all-visible-light-driven multistate chiroptical switch (Fig. 6A). **M2**-**M4** also show visible-light responsiveness up to 470 nm light, similarly to **M1** (Supplementary Figs. 1–3).

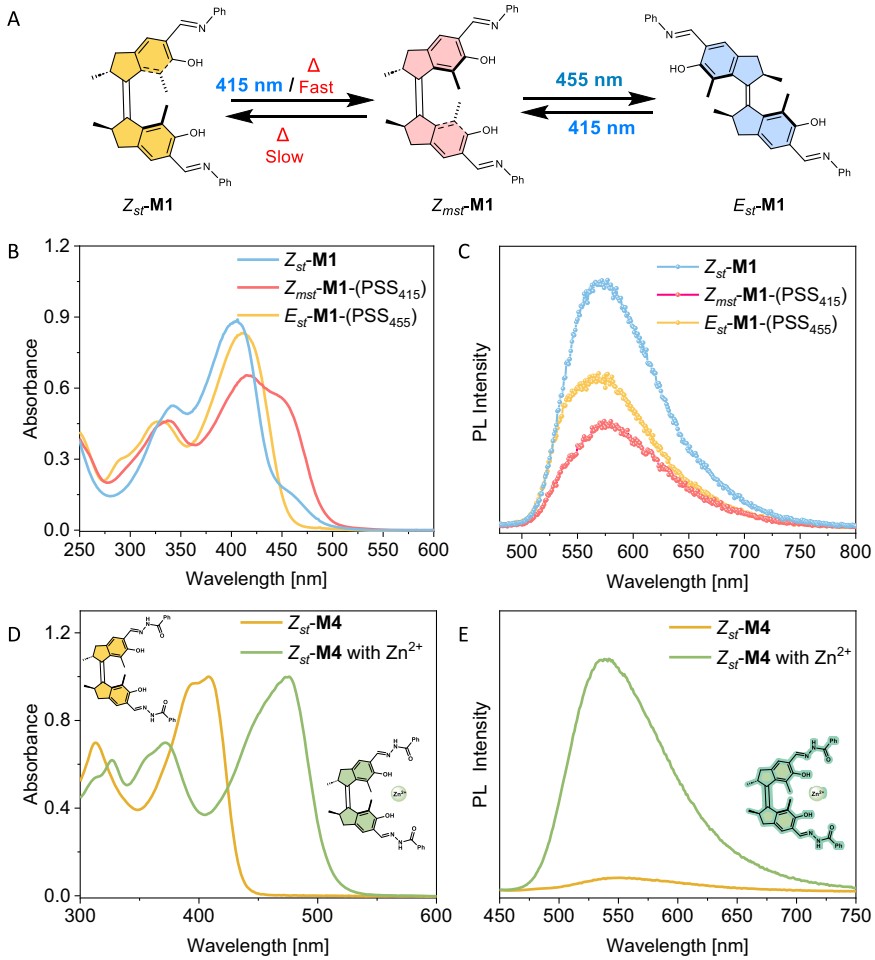

**Fig. 6 | Multistate photoswitching behaviour of molecular motors and their switchable properties. A** Illustration of first-generation molecular motor **M1** as an all-visible-light-driven multistate photoswitch. **B** UV-Vis spectra of **M1** (in *i*-PrOH) in different states by visible-light activation. **C** Modulation of luminescence of **M1** depending on different motor states ($\lambda_{ex}$ = 410 nm). **D** Normalized UV-Vis analysis of binding of **M4** to $Zn^{2+}$ (large access equiv. of Zn(OTf)$_2$ in DMSO). **E** Activation of photoluminescence of **M4** upon $Zn^{2+}$ addition ($\lambda_{ex}$ = 400 nm).

To illustrate the potential of the previously described functional molecular motors, **M1** was employed for the modulation of multistate luminescence. $Z_{st}$-**M1** was found to be a luminescence active compound but no solvent dependent luminescence behaviour was observed (Supplementary Fig. 21). We anticipated that the switching of different states of **M1** could lead to the multistate luminescence activation. To our delight, upon 415 nm light irradiation of $Z_{st}$-**M1**, the PL of the system is significantly reduced (Fig. 6C). Subsequent illumination with 455 nm light, going from $Z_{mst}$-**M1** to $E_{st}$-**M1**, leads to a partial increase of luminescence compared to that of the initial state. The PL of **M1** can be reversibly modulated for several cycles without fatigue, indicating excellent and advantageous (photo)stability (Supplementary Fig. 25). These data show that the intensity of PL within a single molecule is controlled by visible light-induced toggling between the different motor states.

Acylhydrazone compounds are known to bind metal ions, hence we explored whether **M4** could be used to control ion binding. Metal binding has previously been employed in our group to modulate several properties of molecular motors, such as their rotation speed, absorption wavelength and the mechanism of the thermal part of the rotary cycle[28,73,74]. Upon $Zn^{2+}$ addition, a ~ 70 nm redshift in the UV-Vis spectra is observed (Fig. 6D), confirming the strong binding affinity of **M4**. In contrast to $Z_{st}$-**M1**, **M4** show negligible luminescence in the $Z_{st}$-**M4** state. Due to the weak binding affinity of **M4** as a ligand, up to 30 equivalents of $Zn^{2+}$ are required for quantitative binding, which is

consistent with literature precedence (Supplementary Fig. 24)[75]. Surprisingly, after binding of $Zn^{2+}$ to a solution of **M4**, the PL of $Z_{st}$-**M4** is fully turned on, illustrating that metal coordination leads to PL modulation in this system (Fig. 6E). Detailed investigation of the photoisomerization behaviour of the **M4**-Zn complex shows that $Z_{st}$-**M4**-Zn can be quantitatively (PSS$_{455}$ 95:5) converted to $Z_{mst}$-**M4**-Zn as observed by UV-Vis and $^1$H NMR spectroscopy upon irradiation with 455 nm light (Supplementary Fig. 23). Interestingly, the PL of **M4**-Zn is practically quenched upon isomerization to $Z_{mst}$-**M4**-Zn, meanwhile $E_{st}$-**M4**-Zn exhibits similar emission intensity as $Z_{st}$-**M4**-Zn (Supplementary Fig. 22). These results show that simple metal complexation reveals additional features, i.e. switchable photoluminescence, generating a non-trivial dual-functioning molecular motor system[27,36].

In summary, we have established a practical strategy to access a series of versatile visible-light-driven first-generation molecular motors based on the salicylidene Schiff base motif. Using this simple structural modification, redshifts up to 100 nm for the absorption bands of the molecular motors compared to the absorption of the widely used first-generation motor scaffold have been realized, resulting in an all-visible-light-driven function without affecting the motor's unidirectional rotary behaviour. Interestingly, instalment of the salicylidene Schiff bases in both halves of the motor scaffolds renders these motors adaptable to the surroundings, i.e., the photostationary state shows dependency on temperature as well as solvent

polarity, which is attributed to the equilibrium between OH- and NH- forms in the presented design. We foresee that this dependency will provide opportunities for additional control over motor rotation which leads to further fine-tuning of its dynamic behaviour.

Furthermore, extension of this strategy to the structurally related acylhydrazone functionality still permits motor rotation upon visible-light illumination. Additional exploitation of these motor designs shows that they are suitable for implementation in areas such as (light) responsive organic emitters and metal ion sensing. Given the fact that the salicylidene Schiff base motif has been widely exploited in the fields of dynamic and supramolecular chemistry, catalysis and porous materials such as covalent organic frameworks, and in combination with the dynamic chirality of molecular motor cores, we foresee ample opportunity for future development using this motor platform to construct versatile all-visible-light responsive dynamic systems and materials.

## Methods

### Steady-state UV-vis absorption spectroscopy studies

Solutions of Motors were prepared in different organic solvents. 1.0 cm quartz cuvette was used for UV measurements with 2.0 mL solution in it. Samples were placed in the cuvette holder of an Agilent 8453 UV-vis Diode Array System, equipped with a Quantum Northwest Peltier controller. Irradiation studies were carried out using fibre-coupled LEDs (M365F1, M415FP1, M420F2, M455F1, M470F1) obtained from Thorlabs Inc.

### NMR irradiation studies

For isomerization studies, *ca.* $2 \times 10^{-3}$ M solutions of samples were prepared in $CD_2Cl_2$ or DMSO-*d6* solvents. For each experiment 600 µL samples were transferred into an NMR tube that used for in situ irradiation studies. The NMR tube was subsequently fitted with a glass fibre cable. NMR spectra were recorded on a Varian Unity Plus 500 NMR spectrometer. Chemical shifts are given in parts per million (ppm) relative to the residual solvent signal.

## Data availability

Data that support the findings of this study are available within the Supplementary Information. All data are available from the corresponding author upon request. The XYZ coordinates of the DFT-optimized structures are available as source data. Source data are provided with this paper.

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

## Acknowledgements

Financial support from the Netherlands Organization for Scientific Research (NWO-CW), the Dutch Ministry of Education, Culture and Science (Gravitation program No.024.001.035), the China Scholarship Council (CSC PhD Fellowship No.201808330459 to J. S.). We thank Dr. Youxin Fu, Dr. Alexander Ryabchun, Dr. Wojciech Danowski, Dr. Jianyu Zhang and Yahan Shan from University of Groningen for their help to this project and fruitful discussions.

## Author contributions

S.V.V. and J.S. contributed equally to this work. J.S. and B.L.F. conceived the project. B.L.F. guided and supervised the research. S.V.V. and J.S. synthesized compounds, S.V.V. and J.S. led the project, and carried out all experimental studies and characterizations. C.N.S. performed computational studies. All authors discussed the results and commented on the manuscript.

## Competing interests

The authors declare no competing interests.
