## [Peer Review File · Nature Communications]

All-Visible-Light-Driven Salicylidene Schiff-Base-Functionalized Artificial Molecular MotorsREVIEWER COMMENTS

Reviewer #1 (Remarks to the Author):

In this article, Feringa and co-workers described the design, synthesis and studies on overcrowded alkenes functionalized with Schiff's base salicylidene units that can show longer wavelength absorptions around 400 nm, enabling them visible light-driven molecular motors. A range of molecular systems with smart functionalization at the remote positions from the actual machine (olefin) was carefully designed to achieve longer wavelength absorption. Despite using computations to a certain extent in predicting absorption characteristics, the designs were mainly made through a logical extension. All the systems were extensively studied through UV-vis and NMR spectroscopy to understand the involvement of multistate and unidirectional rotation. CD spectroscopy was used for confirmation of the unidirectional rotations. The thermal steps and their barriers were unravelled in the process through spectroscopy. Solvent and temperature effects were also studied to understand the adaptability of the machine towards environmental changes. Furthermore, the photoluminescence property of the salicylidene derivatives was utilized in the light control of it. They have also demonstrated the quenching of photoluminescence by Zn²⁺ metal ions. Thus, the manuscript exemplifies the importance of clever design in undermining the ever-demanding visible-light-driven molecular machines and glorifies their glimpses of application potential. Overall, the manuscript is an important contribution to the field and demands publication in Nature Communications. However, a few points need to be addressed before it gets published.

General:

Line 19: In the abstract "Remarkable redshifts up to 100 nm were.." redshifts "in absorption" can be mentioned.

Line 99: What is Motor S2?

Line 108: In synthesis, which enantiomer can be mentioned.

Line 147: It would be helpful to describe the Zst, Emst, etc., in one of the figures and cite them during the first instance of use. Currently, these were used before describing them in the subsequent text.

Line 148: What is motor S1?

Line 157: What is THI?

Line 162: A structural scheme in the interconversions of these intermediates should be cited or referred to.

Line 166: The notations of Z, Zst and/or Zmst are inconsistent. They should be uniform.

Line 181: The initial diastereoisomer Zmst-M1 via the second THI..should be "Zst-M1".

Line 183: Figure 4E needs to be corrected.

Line 217: lambdamax (spelling)

Line 229: Here, the temperatures shown are rt and lower temperatures. However, the line related to line broadening in ¹H NMR indicates elevated temperatures. Only UV-vis data shows higher temperatures in Figure S13. Also, the states referred to in this line (Zmst-M1) and the one indicated in Figure 3C (Est-M1 in CD₂Cl₂) are different. It should be corrected.

Line 230: This should be Figure 3C.

Several UV-vis data have "absorbance" and "absorbance (a.u.)" scales. They may be made consistent.

Supporting information figures 9-12 should be matched for the compound numbers in the figures with captions. This needs to be confirmed at several places.

Technical:

Line 140: In Table 1, are the computed absorptions of M-OH derivatives related to the hydrogen-bonded or non-hydrogen-bonded structures? Will there be any difference in absorption if the OH group adopts an anti-conformation?

A description of the program, method and type of computations used, and appropriate reasons for such calculations with references must be included.

Table 2: Why M4 has a high thermal half-life compared to others? How does DMSO influence the process?

Figure 2A: The absorbance in the x-axis should be normalized (as mentioned in the caption)

Figure 3A: Did authors attempt IR spectroscopy to obtain insights into the tautomeric structures? Inspecting O-H/C=N and N-H/C=O stretching frequencies can help get information on tautomeric structures. The normalized spectra show some weak absorptions in selected solvents attributed to keto forms. Does computational evidence (appearance of an absorption feature with appropriate oscillator strength) also support this assignment?

Figure 3C: In Figure 3C, Why do the signals around 13 ppm and 2.4 ppm behave differently? An explanation can be added.

Line 222: What wavelength was used in non-polar solvent (in fact, "apolar" solvent should be corrected)? Is the reduction due to the lowering of the absorption cross-section?

Line 223: Do the protic solvents assist in tautomerization or stabilizing the tautomers through additional protonation (through hydrogen bonding)? Polar protic solvent favours keto forms; please refer to J. Am. Chem. Soc. 1977, 99, 3538.

Figure 5B: Is such multistate photoluminescence modulation consistent if repeated over a few cycles?

Figure 5D: There is no description of the binding aspects of Zn²⁺ with M4. Does the motor form a 1:1 or 1:2 complex by binding with Zn ions? A quantitative description of this aspect can be insightful.

Figure 5D: Can photoisomerisation studies be performed after the addition of zinc? Does photoluminescence enhancement associate with the rigidification of the motor / or influence the isomerization upon coordination of salicylidene units? How about testing PL if zinc is added to the motor in any form other than Zst?

Reviewer #2 (Remarks to the Author):

The manuscript "All-visible-light-driven salicylidene Schiff-base-functionalized artificial molecular motors" by van Vliet, Sheng, Stindt and Feringa presents an interesting first-generation molecular motor driven entirely by visible light, based on salicylidene Schiff bases. The demonstrated original system is appealing in a few aspects: first, the majority of previously reported molecular motors are propelled with highly energetic UV light. The current demonstration devoids of these frequencies in favor of visible frequencies, which means decreased tendency for degradation, as well as increases the scope of potential applicability in sensitive systems, particularly of biological nature. While the way towards a nanomachine propelled by light and operational in human blood might seem still long, this report brings an additional important stone to pave it. Another appealing aspect is the environmental sensitivity of the demonstrated system (solvents, temperature) - which is undersdtandable given the known dynamic nature of Schiff bases and its tautomerism. As correctly noted, it might provide another dimension for controlling the behavior of the demonstrated system. Finally, the authors bring immediate practical aspects to the demonstrated scaffold, by showing the propensity of the motor M1 for photomodulation of its luminescence. Finally, the motor M4 strongly increases in luminescence upon binding with zinc ion.

The work is definitely interesting and innovative. The experiments in my oppinion sufficiently support the conclusions, the methodology is sound, and sufficient data is provided for reproducibility. I recommend publication upon the minor issues listed below are addressed:

1) the use of visible light for photoswitchable systems is generally desirable i.a. for the stability reasons. Could the authors provide in the manuscript a paragraph discussing the stability of the particular motors presented in their work upon multiple rotation cycles (I hope I did not ommit this information in the manuscript), maybe in comparison with other motors propelled with UV light (the first generation, and maybe the other ones) - to clearly prove (or disprove) the stability advantage in this particular case

2) binding of zinc ions to the motor M4 (Figure 5d-e) - could the authors comment more broadly (a paragraph) on the behavior of the complex formed between the M4 and zinc ion - can this still be

switched/rotated with light (with eventual detachment from the metal ion?)? Or does the presence of the metal ion fully inhibits the rotation, and changes the motor into a luminescent ion sensor?

Reviewer #3 (Remarks to the Author):

This paper reports on the synthesis and characterization of first-generation rotary overcrowded-alkene motors made responsive to visible light by the incorporation of a salicylidene Schiff-base motif. Besides describing the synthesis and spectroscopic measurements probing the actual motor function, the paper investigates solvent effects on the motor function that come into play by influencing the keto-enol equilibrium of the salicylidene Schiff base. Moreover, preliminary results highlighting potential applications of the motors (control of photoluminescence and ion binding) are described, as are results from computational studies that help identify the structural characteristics of the motors that are needed to bring their light absorption into the visible regime.

The paper is well written and the results are likely to be of interest to a broad readership. However, there are a number of important issues that I believe should be addressed in a revised manuscript, before this work can be accepted for publication:

(1) Part of the motivation for the work, as presented on page 2, is that all previous efforts to make overcrowded-alkene motors responsive to visible light are based on second-generation motors. Accordingly, it is argued that this fact makes it worthwhile to realize the same goal based instead on first-generation motors. However, the focus on first-generation motors also seems to bring back some of the problems that are more pronounced for these motors, than for subsequent second-generation motors. One such problem is the large magnitude of the barriers for the thermal helix inversion steps. Indeed, as expected, these barriers are found to be very large for the current motors, amounting to at least 100 kJ/mol (see Table 2). Thus, these motors are not particularly potent, and it is surprising that this deficiency is not acknowledged (let alone discussed) in the manuscript.

(2) It seems likely that the Z/E photoisomerizations that produce the rotary motion become less dynamically favorable upon introduction of a salicylidene Schiff-base motif, as this motif opens up two alternative photochemical reaction channels that compete with the Z/E photoisomerizations: excited-state proton transfer and Z/E photoisomerization of the "imine/amine" functionality. This issue is to some extent addressed by the results in Figure 3, but I think that this is a weak point that should be acknowledged rather than presented as something positive on page 8: "... these findings might facilitate the exploration of smart, adaptable motor-based systems that can adjust to the conditions of their environment". Indeed, I believe it is fair to say that overcrowded-alkene motors

have generally rather poor quantum yields. Thus, modifications that are likely to make the quantum yields worse should be acknowledged as such.

(3) Finally, it is unfortunate that this work is presented (on page 3) as “taking advantage of our recently developed salicylaldehyde motor compounds”, but that the corresponding paper (ref. 53) remains a manuscript in preparation, to which readers have no access.

Detailed response.

REVIEWER COMMENTS

Reviewer #1 (Remarks to the Author):

In this article, Feringa and co-workers described the design, synthesis and studies on overcrowded alkenes functionalized with Schiff's base salicylidene units that can show longer wavelength absorptions around 400 nm, enabling them visible light-driven molecular motors. A range of molecular systems with smart functionalization at the remote positions from the actual machine (olefin) was carefully designed to achieve longer wavelength absorption. Despite using computations to a certain extent in predicting absorption characteristics, the designs were mainly made through a logical extension. All the systems were extensively studied through UV-vis and NMR spectroscopy to understand the involvement of multistate and unidirectional rotation. CD spectroscopy was used for confirmation of the unidirectional rotations. The thermal steps and their barriers were unraveled in the process through spectroscopy. Solvent and temperature effects were also studied to understand the adaptability of the machine towards environmental changes. Furthermore, the photoluminescence property of the salicylidene derivatives was utilized in the light control of it. They have also demonstrated the quenching of photoluminescence by Zn^{2+} metal ions. Thus, the manuscript exemplifies the importance of clever design in undermining the ever-demanding visible-light-driven molecular machines and glorifies their glimpses of application potential. Overall, the manuscript is an important contribution to the field and demands publication in Nature Communications. However, a few points need to be addressed before it gets published.

We thank the referee for the positive response, scholarly evaluation and valuable suggestions to clarify some points.

General:

Line 19: In the abstract "Remarkable redshifts up to 100 nm were.." redshifts "in absorption" can be mentioned.

We have amended line 19 and added "in absorption" to the Abstract.

Line 99: What is Motor S2?

We have renamed Motor S2 to $Z_{SF}M1-OMe$ in the main text for clarification and we refer the reader to "(Supplementary Information, section 1)" for synthetic details and characterization data.

Line 108: In synthesis, which enantiomer can be mentioned.

We have added the appropriate "(R,R)-(P,P)" descriptor in the revised manuscript to explain that enantiopure parent motor has been used in its synthesis.

Line 147: It would be helpful to describe the *Zst*, *Emst*, etc., in one of the figures and cite them during the first instance of use. Currently, these were used before describing them in the subsequent text.

We apologize for the initial lack of clarity and have now described all the motor isomers with stereochemical details using proper subscripts and explanations thereof in parentheses at the first instance of use:

***Z_{st}*-M1** (*cis* stable isomer, *i*-PrOH, 293 K)

***E_{mst}*-M1** (*trans* metastable isomer)

***Z_{mst}*-M1** (*cis* metastable isomer)

***E_{st}*-M1** (*trans* stable isomer)

Line 148: What is motor S1?

We have now referred the reader to “(Figure 2A)” for the actual structure of **Motor-S1**.

Line 157: What is THI?

The thermal helix inversion (THI) process is described in Line 43.

Line 162: A structural scheme in the interconversions of these intermediates should be cited or referred to.

We have referred the reader to “(Figure 2B and Supplementary Figure 9)” in the main text to give the reader structural insight in the 360° rotary cycle and the corresponding isomerizations / interconversions taking place.

Line 166: The notations of *Z*, *Zst* and/or *Zmst* are inconsistent. They should be uniform.

We apologize for the inconsistency and have now amended the subscript notations in the revised manuscript.

Line 181: The initial diastereoisomer *Zmst*-M1 via the second THI..should be “*Zst*-M1”.

We have corrected the mistake into “***Z_{st}*-M1**”.

Line 183: Figure 4E needs to be corrected.

We have corrected it into “**Figure 4D**” in our revised manuscript.

Line 217: *lambdamax* (spelling)

We have amended the spelling and highlighted **λ_{\max}** in the main text.

Line 229: Here, the temperatures shown are *rt* and lower temperatures. However, the line related to line broadening in ¹H NMR indicates elevated temperatures. Only UV-vis data shows higher

temperatures in Figure S13. Also, the states referred to in this line (Zmst-M1) and the one indicated in Figure 3C (Est-M1 in CD₂Cl₂) are different. It should be corrected.

We thank the reviewer for pointing out this inconsistency and have now amended caption C of Figure 3 and highlighted the changes made:

Caption C. Chemical shifts of OH resonance in ¹H spectra of E_{sr}-M1 in CD₂Cl₂ at elevated temperatures, indicating the decreasing strength of H-bonding (from top to bottom).

Besides, it is noteworthy to mention that deuterated dichloromethane has been used during ¹H NMR studies. Due to the relatively low boiling point of CD₂Cl₂, the temperature could only be elevated to approximately 30 °C during these studies to prevent undesired solvent evaporation.

Line 230: This should be Figure 3C. Several UV-vis data have "absorbance" and "absorbance (a.u.)" scales. They may be made consistent.

It has been corrected into "Figure 3C" in the revised manuscript. We did realize absorbance is a dimensionless quantity, so the use of arbitrary units [a.u.] is incorrect and we have corrected this in our revised spectra. Furthermore, we have added additional captions to distinguish between normalized spectra and non-normalized spectra.

Supporting information figures 9-12 should be matched for the compound numbers in the figures with captions. This needs to be confirmed at several places.

We thank the reviewer for bringing this to our attention. In our revised manuscript, we have corrected and double checked all the figures for the sake of consistency.

Technical:

Line 140: In Table 1, are the computed absorptions of M-OH derivatives related to the hydrogen-bonded or non-hydrogen-bonded structures? Will there be any difference in absorption if the OH group adopts an anti-conformation?

The structures represent the hydrogen-bonded structures. To clarify this, we now explicitly mention it in the text. The hydrogen bonded structures constituted the preferred conformations in the conformers generated by our conformer search. To further probe the effect of the hydrogen bond, we also calculated the structures M-OMe (see Table 1), which feature a methoxy substituent instead of a hydroxy substituent. This substituent has nearly identical electronic properties, but is not capable of forming a hydrogen bond. We therefore infer that MeO-substituted structures have approximately the same properties as OH-substituted structures in non-hydrogen-bonded conformations. As can be seen from Table 1, the M-OMe structures have indeed more blue-shifted maximum absorption wavelengths than their M-OH analogues.

A description of the program, method and type of computations used, and appropriate reasons for such calculations with references must be included.

A description of the type of calculation and the level of theory is provided in the revised main text and in the supporting information. A more elaborate description is provided in the supporting information, which also provides details on the programs used and relevant parameters. Additionally, we have added references to the computational methods in the main text as well as a sentence explaining the use of TDDFT calculations:

“Time-dependent density functional theory (TDDFT) is a well-established extension to density functional theory (DFT) to calculate properties for time-dependent systems, such as excitation energies and photoabsorption spectra.”

Table 2: Why M4 has a high thermal half-life compared to others? How does DMSO influence the process?

We thank the reviewer for the opportunity to explain this matter in more detail. We used DMSO as the solvent in choice for the Eyring analysis of M4 due to its poor intrinsic solubility in a wide range of solvents, e.g. *i*-PrOH, CH₂Cl₂ or acetonitrile.

The Thermal Helix Inversion (THI) is a complex thermal unimolecular reaction which is influenced by a vast number of variables comprised of various interactions between solvent and solute and solvent molecules among themselves (*Phys. Chem. Chem. Phys.*, 2016, **18**, 26725). Viscosity has been established as one of the key parameters effecting this thermal process; the half-life of an identical molecular motor differs significantly in different solvent systems having varying viscosities (*Faraday Discuss.*, 2009, **143**, 319).

Furthermore, the solvent shell around a molecular motor needs to be rearranged upon isomerization. This restructuring comprises a complex interplay between different non-covalent interactions, e.g. H-bonding, van der Waals and dipole-dipole interactions, among the solvent and the molecular motor (*Chemistry*, 2022, **4**, 185). Hence, due to the complex nature of the THI's rate dependency, a comparison between the half-lives of different molecular motors amongst different solvent systems cannot unambiguously be made. DMSO undoubtedly has an influence on the THI and the corresponding half-life of M4, however pointing out its exact impact remains elusive.

Figure 2A: The absorbance in the x-axis should be normalized (as mentioned in the caption)

We have modified Figure 2A accordingly.

Figure 3A: Did authors attempt IR spectroscopy to obtain insights into the tautomeric structures? Inspecting O-H/C=N and N-H/C=O stretching frequencies can help get information on tautomeric structures. The normalized spectra show some weak absorptions in selected solvents attributed to keto forms. Does computational evidence (appearance of an absorption feature with appropriate oscillator strength) also support this assignment?

We thank the reviewer for this scholarly suggestion and we have tried to obtain information regarding the stretching frequencies of interest from the IR spectra of solutions in CH₂Cl₂ and MeOH in transmission mode and ATR mode. However, the low solubility in MeOH and fast evaporation of CH₂Cl₂ solutions prevented the acquisition of high-quality spectra.

Computational support for the assignment to the keto form is provided in Table S1 in the supporting information. Here it is shown that the absorption wavelength corresponding to the first absorption of these tautomers for motor **M1-M3** is ~40 nm redshifted compared with the enol tautomers. Regarding the oscillator strength, we have added a sentence to the caption of Table S1, indicating that these transitions show the strongest oscillator strength in all cases.

Figure 3C: In Figure 3C, why do the signals around 13 ppm and 2.4 ppm behave differently? An explanation can be added.

The shift of the active hydroxy proton (OH) at δ 13.7 ppm at elevated temperature is related to the decreasing strength of H-bonding with the imino-nitrogen atom. However, the resonance corresponding to the methyl substituent (CH₃) on the aryl ring at δ 2.4 ppm is irrelevant to this H-bonding, which mainly correlates to the motor structure itself.

Furthermore, we observed a modest shift at δ 3.0 and δ 2.8 ppm, corresponding to the protons attached to motor's five-membered cyclopentane motif, thereby showing a similar trend with the methyl resonance at δ 2.4 ppm which all correlate to the motor core structure. Worth mentioning is that similar differences in chemical shifts, i.e. temperature difference in chemical shifts belonging to proton resonances not actively involved in the H-bonding, have been observed by Wei-Hong Zhu and colleagues in their work on intramolecular proton transfer in visible-light activated dithienylethenes photoswitches (*J. Am. Chem. Soc.* 2019, **141**, 18467. SI, section 3). For clarification we have rephased caption 3C as follows:

“Caption C. Chemical shifts of OH resonance in ¹H spectra of *E_{st}*-**M1** in CD₂Cl₂ at elevated temperatures, indicating the decreasing strength of H-bonding (from top to bottom).”

Line 222: What wavelength was used in non-polar solvent (in fact, "apolar" solvent should be corrected)? Is the reduction due to the lowering of the absorption cross-section?

For our irradiation studies in a range of solvents (Supporting Information, section 4), 415 nm has been the wavelength of choice corresponding to the *Z_{st}* → *Z_{mst}* isomerization. We assume the reduction of PSS is ascribed to lowering of the absorption cross-section as the process of tautomerization has significantly been impacted in non-polar solvents. We have also corrected “apolar” to “non-polar” in the manuscript.

Line 223: Do the protic solvents assist in tautomerization or stabilizing the tautomers through additional protonation (through hydrogen bonding)? Polar protic solvent favours keto forms; please refer to J. Am. Chem. Soc. 1977, 99, 3538.

We are grateful for the additional literature and have added an additional reference (68) to our revised manuscript.

Protic solvents (as lowering temperature) will assist in stabilizing the tautomer of its keto form, which aids the photoisomerization of the motor in our case. Interestingly, similar results were observed by Wei-Hong Zhu and co-workers in their study on dithienylethenes photoswitches as referred to in citation 53 (*J. Am. Chem. Soc.* 2019, **141**, 18467)

Figure 5B: Is such multistate photoluminescence modulation consistent if repeated over a few cycles?

To study the fatigue resistance of the photoluminescence (PL) modulation, we performed five consecutive switching cycles. The PL fatigue study started from the isomerization of $Z_{st}\text{-M1}$ to $Z_{mst}\text{-M1}$ using 405 nm light, whereafter switching between $Z_{mst}\text{-M1}$ (PSS₄₀₅) and $E_{st}\text{-M1}$ (PSS₄₆₀) was performed for successive five cycles. After cycling, leaving the sample in the dark at elevated temperature allowed $Z_{mst}\text{-M1}$ to undergo the corresponding THI to reobtain $Z_{st}\text{-M1}$.

These results show robust switching of PL between the three states, namely $Z_{st}\text{-M1}$, $Z_{mst}\text{-M1}$ and $E_{st}\text{-M1}$. Note that the final PL intensity, after the occurrence of THI, is lower than the initial PL due to the photostationary state (PSS) distribution in which both $Z_{st}\text{-M1}$ and $E_{st}\text{-M1}$ are present. We have added additional data as in Figure 25 in the Supplementary Information.

Supplementary Figure 25. a) PL Fatigue study starting from $Z_{st}\text{-M1}$ to $Z_{mst}\text{-M1}$, then switching between $Z_{mst}\text{-M1}$ (PSS₄₀₅) and $E_{st}\text{-M1}$ (PSS₄₆₀) for five cycles, and finally heating back to $Z_{st}\text{-M1}$. and b) PL spectra of initial $Z_{st}\text{-M1}$, $Z_{mst}\text{-M1}$ (PSS₄₀₅) and $E_{st}\text{-M1}$ (PSS₄₆₀).

Figure 5D: There is no description of the binding aspects of Zn²⁺ with M4. Does the motor form a 1:1 or 1:2 complex by binding with Zn ions? A quantitative description of this aspect can be insightful.

Due to the weak binding affinity of **M4** as ligand, over 30 equivalents of Zn²⁺ are required in order to get quantitative binding, as observed by UV-Vis titration curves. These findings are consistent with literature precedence on a similar acyl hydrazone ligand requiring (large) excesses of [Zn²⁺] to achieve quantitative binding (*Adv. Mater.* 2021, **33**, 2105113).

Noteworthy to mention is that we have attempted MALDI-TOF experiments to gain insight in the binding stoichiometry of **M4**, however this led to varying and inconclusive results. Besides, attempts to grow crystals suitable for single crystal X-ray diffraction via a wide variety of approaches to get insight in the solid-state structure of **M4-Zn** have proven to be unfruitful. We have added a new sentence in the in our revised manuscript commenting on the binding aspects of **M4**:

“Due to the weak binding affinity of **M4** as a ligand, up to 30 equivalents of Zn²⁺ are required for quantitative binding, which is consistent with literature precedence (Supplementary Figure 24).”

Supplementary Figure 24. Titration of a) $Z_{st}\text{-M4}$ and b) $Z_{mst}\text{-M4}$ with Zn(NO₃)₂ in THF monitored by UV-Vis spectroscopy.

Figure 5D: Can photoisomerisation studies be performed after the addition of zinc? Does photoluminescence enhancement associate with the rigidification of the motor / or influence the isomerization upon coordination of salicylidene units? How about testing PL if zinc is added to the motor in any form other than Z_{st}?

We have performed photoisomerization studies of **M4** after Zn²⁺ addition by UV-Vis and ¹H NMR spectroscopy in DMSO. Surprisingly, we observed clean and nearly quantitative photoisomerization (PSS₄₅₅ 95:5) from the $Z_{st}\text{-M4-Zn}$ complex to the $Z_{mst}\text{-M4-Zn}$ complex. Interestingly, besides drastically enhancing the PL of the system, the addition of Zn²⁺ enables

photoisomerization upon irradiation with 455 nm wavelength of light, thereby red shifting the excitation wavelength of **M4** by at least 40 nm. This effectuated dual-functioning, namely PL and visible-light activated photoswitching, of $Z_{st}\text{-M4-Zn}$ shows the future potential of these kinds of motor systems. In this regard, we have added this additional data as Figure 23 to the Supplementary Information.

Supplementary Figure 23. Photoisomerization of **M4** with Zn^{2+} monitored by a. UV-Vis and b. 1H NMR spectroscopy.

Furthermore, photomodulation of the PL intensity for the **M4-Zn** complex was observed. Specifically, the $Z_{mst}\text{-M4-Zn}$ complex shows negligible PL compared to $Z_{st}\text{-M4-Zn}$ complex. Besides, we measured the PL of the $E_{st}\text{-M4-Zn}$ complex, which resulted in similar emission intensity compared to $Z_{st}\text{-M4-Zn}$ albeit with a modest blue-shift. Irradiation of $E_{st}\text{-M4-Zn}$ using 455 nm resulted in the isomerization towards $Z_{st}\text{-M4-Zn}$ complex accompanied with a sharp decrease in PL emission, consistent with the aforementioned results obtained for the $Z_{mst}\text{-M4-Zn}$ PL studies. We have added these new results as Figure 22 to the revised Supplementary Information. In our revised manuscript, we dedicate an additional paragraph to our new findings:

Detailed investigation of the photoisomerization behaviour of the **M4-Zn** complex shows that $Z_{st}\text{-M4-Zn}$ can be quantitatively (PSS₄₅₅ 95:5) converted to $Z_{mst}\text{-M4-Zn}$ as observed by UV-Vis and 1H NMR spectroscopy upon irradiation with 455 nm light (Supplementary Figure 23). Interestingly, the PL of **M4-Zn** is practically quenched upon isomerization to $Z_{mst}\text{-M4-Zn}$, meanwhile $E_{st}\text{-M4-Zn}$ exhibits similar emission intensity as $Z_{st}\text{-M4-Zn}$ (Supplementary Figure 22). These results show that simple metal complexation untaps novel features, i.e. switchable photoluminescence, generating a non-trivial dual-functioning molecular motor system.

We propose Zn^{2+} coordinates to the flanking acyl hydrazone units imparting enhanced rigidity to these motifs while not significantly interfering with the central motor axle, thereby maintaining motor isomerization. Nevertheless, probing the exact conformational behavior of these **M4-Zn** complexes in solution is non-trivial. In-depth analysis of the structural changes as well as the exact stoichiometry of these motor-metal complexes is currently an ongoing line of research in our laboratories.

Supplementary Figure 22. Photoluminescence of **M4** with and without Zn^{2+} at different states ($\lambda_{ex} = 400$ nm).

Reviewer #2 (Remarks to the Author):

The manuscript "All-visible-light-driven salicylidene Schiff-base-functionalized artificial molecular motors" by van Vliet, Sheng, Stindt and Feringa presents an interesting first-generation molecular motor driven entirely by visible light, based on salicylidene Schiff bases. The demonstrated original system is appealing in a few aspects: first, the majority of previously reported molecular motors are propelled with highly energetic UV light. The current demonstration devoids of these frequencies in favor of visible frequencies, which means decreased tendency for degradation, as well as increases the scope of potential applicability in sensitive systems, particularly of biological nature. While the way towards a nanomachine propelled by light and operational in human blood might seem still long, this report brings an additional important stone to pave it. Another appealing aspect is the environmental sensitivity of the demonstrated system (solvents, temperature) - which is understandable given the known dynamic nature of Schiff bases and its tautomerism. As correctly noted, it might provide another dimension for controlling the behavior of the demonstrated system. Finally, the authors bring immediate practical aspects to the demonstrated scaffold, by showing the propensity of the motor M1 for photomodulation of its luminescence. Finally, the motor M4 strongly increases in luminescence upon binding with zinc ion. The work is definitely interesting and innovative. The experiments in my opinion sufficiently support the conclusions, the methodology is sound, and sufficient data is provided for reproducibility. I recommend publication upon the minor issues listed below are addressed:

We thank the referee for the positive response acknowledging the innovative aspects of our findings. We greatly appreciate the scholarly evaluation and valuable suggestions.

1) the use of visible light for photoswitchable systems is generally desirable i.e. for the stability reasons. Could the authors provide in the manuscript a paragraph discussing the stability of the particular motors presented in their work upon multiple rotation cycles (I hope I did not ommit this information in the manuscript), maybe in comparison with other motors propelled with UV light (the first generation, and maybe the other ones) - to clearly prove (or disprove) the stability advantage in this particular case

We share the view of the reviewer regarding the stability improvement of visible light activated molecular motors or switches. The vast majority of first-generation molecular motors use high energetic UV-light, about 312 nm, as excitation wavelengths, thereby hampering their potential in the areas of soft materials and biology. Visible light responsiveness will drastically broaden the applicability of these motor candidates in dynamic molecular systems due to better penetration depth of benign light in bulk materials, a task which has not yet been realized prior to this study by first-generation molecular motors.

To assess stability, we have performed multiple irradiation and consecutive heating cycles of **M4** followed by ¹H NMR. Remarkably, after five rounds of rotation cycles **M4** continues to show clean and robust (photo)isomerization in absence of degradation. We have added this data as Figure 26 in the revised Supplementary Information.

Additionally, we have performed fatigue studies on **M1** by probing its PL intensity upon multiple irradiation cycles (vide supra). These results show retention of PL intensity over the course of at least five consecutive switching cycles, confirming the high stability on robustness of the system.

To emphasize the importance of fatigue resistance and overall stability, we have added two additional sentences to our revised manuscript:

To demonstrate fatigue resistance and overall stability, **M4** was subjected to multiple consecutive irradiation and heating cycles which were followed by ^1H NMR spectroscopy. Over the course of five rotary cycles, the initial and final spectra of $Z_{\text{st}}\text{-M4}$ showed to be indistinguishable, confirming clean (photo)isomerization and general robustness of this motor system (Supplementary Figure 26). The PL of **M1** can be reversibly modulated for several cycles without observance of fatigue, indicating excellent and advantageous (photo)stability (Supplementary Figure 25).

Supplementary Figure 26. ^1H NMR Fatigue study starting from $Z_{\text{st}}\text{-M4}$ to $Z_{\text{mst}}\text{-M4}$ (PSS₄₀₅), and heating back to $Z_{\text{st}}\text{-M4}$ for 5 cycles.

Supplementary Figure 25. a) PL Fatigue study starting from $Z_{\text{st}}\text{-M1}$ to $Z_{\text{mst}}\text{-M1}$, then switching between $Z_{\text{mst}}\text{-M1}$ (PSS₄₀₅) and $E_{\text{st}}\text{-M1}$ (PSS₄₆₀) for five cycles, and finally heating back to $Z_{\text{st}}\text{-M1}$. and b) PL spectra of initial $Z_{\text{st}}\text{-M1}$, $Z_{\text{mst}}\text{-M1}$ (PSS₄₀₅) and $E_{\text{st}}\text{-M1}$ (PSS₄₆₀).

2) *binding of zinc ions to the motor M4 (Figure 5d-e) - could the authors comment more broadly (a paragraph) on the behavior of the complex formed between the M4 and zinc ion - can this still be switched/rotated with light (with eventual detachment from the metal ion)? Or does the presence of the metal ion fully inhibit the rotation and changes the motor into a luminescent ion sensor?*

We thank the reviewer for giving us the possibility to elaborate on the switching behavior of the **M4-Zn** complex. We have performed additional experiments and dedicated an additional paragraph to our new findings:

Detailed investigation of the photoisomerization behaviour of the **M4-Zn** complex shows that $Z_{\text{sr}}\text{-M4-Zn}$ can be quantitatively (PSS₄₅₅ 95:5) converted to $Z_{\text{mst}}\text{-M4-Zn}$ as observed by UV-Vis and ^1H NMR spectroscopy (Supplementary Figure 23) upon irradiation with 455 nm light. Interestingly, the PL of **M4-Zn** is practically quenched upon isomerization to $Z_{\text{mst}}\text{-M4-Zn}$, meanwhile $E_{\text{sr}}\text{-M4-Zn}$ exhibits similar emission intensity as $Z_{\text{sr}}\text{-M4-Zn}$ (Supplementary Figure 22). These results show that simple metal complexation reveals novel features, i.e. switchable photoluminescence, generating a non-trivial dual-functioning molecular motor system.

Reviewer #3 (Remarks to the Author):

This paper reports on the synthesis and characterization of first-generation rotary overcrowded-alkene motors made responsive to visible light by the incorporation of a salicylidene Schiff-base motif. Besides describing the synthesis and spectroscopic measurements probing the actual motor function, the paper investigates solvent effects on the motor function that come into play by influencing the keto-enol equilibrium of the salicylidene Schiff base. Moreover, preliminary results highlighting potential applications of the motors (control of photoluminescence and ion binding) are described, as are results from computational studies that help identify the structural characteristics of the motors that are needed to bring their light absorption into the visible regime.

The paper is well written and the results are likely to be of interest to a broad readership. However, there are a number of important issues that I believe should be addressed in a revised manuscript, before this work can be accepted for publication:

We thank the referee for the appreciation of our work highlighting the importance for potential applications and noting that our manuscript is of high interest to a broad readership.

(1) Part of the motivation for the work, as presented on page 2, is that all previous efforts to make overcrowded-alkene motors responsive to visible light are based on second-generation motors. Accordingly, it is argued that this fact makes it worthwhile to realize the same goal based instead on first-generation motors. However, the focus on first-generation motors also seems to bring back some of the problems that are more pronounced for these motors, than for subsequent second-generation motors. One such problem is the large magnitude of the barriers for the thermal helix inversion steps. Indeed, as expected, these barriers are found to be very large for the current motors, amounting to at least 100 kJ/mol (see Table 2). Thus, these motors are not particularly potent, and it is surprising that this deficiency is not acknowledged (let alone discussed) in the manuscript.

We agree with the reviewer that the second thermal helix inversion (THI) step of first-generation molecular motors, i.e. from Z_{mst} to Z_{st} , is much slower compared to THI of the second-generation motors due to the significant change in steric hindrance experienced by the involved metastable isomers. In this regard, the use of first-generation motors for continuous rotation applications, in which the motor should rotate with velocious speed, is not ideal yet but it should be noted that there is ample opportunity to adapt the structures (based on the motor program in our group shown in the past decade) to lower barriers and increase the speed for specific application. The current findings solved at least one of the key issues shifting to visible wavelengths. Realize that we have used these first-generation motors to enable continuous rotational motion with Liquid Crystal materials but also use them extremely successful for multistate unidirectional chiral catalysts. Nevertheless, with the shifting focus in contemporary chemistry from static molecules to responsive dynamic systems, there is a growing demand for adaptive multistage molecular switching systems capable of adopting multiple, i.e. more than two, distinctive states. In this respect, first-generation motors could fulfill a pivotal role in the development of such systems as three-state, i.e. Z_{st} , E_{st} and Z_{mst} , chiroptical switches with intrinsic non-destructive readout.

Taking advantage of the sufficiently high thermal barrier of the second THI step, the Z_{mst} isomer serves as a practically stable third state at room temperature, a feature which the second-generation molecular motors lack. To this end, first-generation molecular motors are privileged and potent candidates for the realization of dynamic, multistage switching networks. We have

shown applications of these motors as multistate chiroptical photoswitches which are emphasized in the last section of our manuscript (Demonstration of potential applications as multistate photoswitches driven by visible light). Additionally, our new system shows an additional advantage by means of an extra readout option in the form of modular photoluminescence intensity.

We are convinced that the novel features of our discussed motor systems will further reignite the interest in first-generation motors and concomitantly accelerate the construction of smart and adaptive materials. To elaborate on this, we have added an additional sentence to our revised manuscript:

Whereas second-generation MMs are more suitable for continuous rotation applications, requiring rotary motion at velocious speed, first-generation MMs could fulfil a pivotal role in the growing demand for stimuli-adaptive multistage systems, capable of adopting multiple distinctive states with sequence specificity such as three-state chiroptical switches.

(2) It seems likely that the Z/E photoisomerizations that produce the rotary motion become less dynamically favorable upon introduction of a salicylidene Schiff-base motif, as this motif opens up two alternative photochemical reaction channels that compete with the Z/E photoisomerizations: excited-state proton transfer and Z/E photoisomerization of the “imine/amine” functionality. This issue is to some extent addressed by the results in Figure 3, but I think that this is a weak point that should be acknowledged rather than presented as something positive on page 8: “... these findings might facilitate the exploration of smart, adaptable motor-based systems that can adjust to the conditions of their environment”. Indeed, I believe it is fair to say that overcrowded-alkene motors have generally rather poor quantum yields. Thus, modifications that are likely to make the quantum yields worse should be acknowledged as such.

We can only partially share the concern raised by reviewer 3 about introducing competing photochemical pathways upon incorporation of salicylidene Schiff-base motifs. Indeed, the original light-driven molecular motors that were designed several years ago featured rather low quantum yields for isomerization, especially the second-generation motors. However, quantum yields have been improved over the years and the bis-aldehyde parent motor Z_{st} -**PM** we use in our studies as platform for the construction of visible light-activated first-generation motors possess amazing quantum yields over 80% for photoisomerization (Nat. Chem. 2024, DOI:10.1038/s41557-024-01521-0). We are therefore confident that it is justified to sacrifice some of this high quantum yield of the parent motor in a balancing fashion to an extent in which the photostationary state (PSS) distribution remains largely uncompromised. In this case, potential reduction of the photoisomerization quantum yield is outweighed by the highly advantageous visible light-responsiveness and additional features of our designed first-generation motors. We have incorporated an additional paragraph describing the possible competing photochemical pathways and the potential decrease of the photoisomerization quantum yield accordingly:

“The incorporation of salicylidene Schiff-base moieties in these motor systems introduces the possibility of competing photochemical pathways, i.e., intramolecular excited-state proton transfer⁶⁶ and photoinduced *E-Z* switching of the imine functionality,⁶⁷ which could impart a reduction in the quantum yield of photochemical motor isomerization. However, the benefit that these motors possess, e.g., their visible light responsiveness, outweighs the potential lowering of the photoisomerization quantum yields providing that the photostationary state (PSS) distributions and therefore the corresponding rotary functions remain largely unaffected. With respect to the latter, **M1** and **M2** reach PSS ratios well-over 80% at room temperature in non-polar solvents,

fulfilling the key criteria, and unveil minimal effects of competing photochemical reaction channels for these specific motors.”

Nevertheless, the presence of photochemical competing pathways in molecular motors does not have to be considered as negative per se. Recently, photoluminescent dyes were attached onto molecular motor cores to realize dual function, i.e. unidirectional rotary motion and luminescence.

(3) Finally, it is unfortunate that this work is presented (on page 3) as “taking advantage of our recently developed salicylaldehyde motor compounds”, but that the corresponding paper (ref. 53) remains a manuscript in preparation, to which readers have no access.

Our recently developed salicylaldehyde motor has just been accepted for publication and has appeared online last week. We have included the required details in reference 54 (Nat. Chem. 2024, DOI:10.1038/s41557-024-01521-0).

REVIEWERS' COMMENTS

Reviewer #1 (Remarks to the Author):

The authors have adequately addressed all the concerns in the manuscript. Now, the manuscript is suitable for publication. Congratulations to all the authors!

Reviewer #2 (Remarks to the Author):

The comments raised in the previous round of review have been successfully addressed by the authors - both regarding the stability of the presented motors, and the behavior of motor M4 with bound Zn ion (resulting in switchable photoluminescence).

Thus I recommend publication of the current version of the manuscript without further changes.

Reviewer #3 (Remarks to the Author):

I am happy with the responses by the authors to my comments on the original manuscript, and with the changes made in revised manuscript. Publication of this nice work is therefore recommended.

Bo Durbeej